## Can implementation failure or intervention failure explain the result of the 3D multimorbidity trial in general practice: mixed-methods process evaluation

Cindy Mann ,[1] Ali R G Shaw,[1] Bruce Guthrie,[2] Lesley Wye,[1] Mei-See Man ,[1] Katherine Chaplin ,[1] Chris Salisbury [1]

¹Centre for Academic Primary Care, School of Population Health Sciences, University of Bristol, Bristol, UK
²Division of Population Health Sciences, University of Dundee, Dundee, UK

**Correspondence to**
Dr Cindy Mann;
cindy.mann@bristol.ac.uk

## ABSTRACT

**Objectives** During a cluster randomised trial, (the 3D study) of an intervention enacting recommended care for people with multimorbidity, including continuity of care and comprehensive biennial reviews, we examined implementation fidelity to interpret the trial outcome and inform future implementation decisions.

**Design** Mixed-methods process evaluation using cross-trial data and a sample of practices, clinicians, administrators and patients. Interviews, focus groups and review observations were analysed thematically and integrated with quantitative data about implementation. Analysis was blind to trial outcomes and examined context, intervention adoption, reach and maintenance, and delivery of reviews to patients.

**Setting** Thirty-three UK general practices in three areas.

**Participants** The trial included 1546 people with multimorbidity. 11 general practitioners, 14 nurses, 7 administrators and 38 patients from 9 of 16 intervention practices were sampled for an interview.

**Results** Staff loss, practice size and different administrative strategies influenced implementation fidelity. Practices with whole administrative team involvement and good alignment between the intervention and usual care generally implemented better. Fewer reviews than intended were delivered (49% of patients receiving both intended reviews, 30% partially reviewed). In completed reviews >90% of intended components were delivered, but review observations and interviews with patients and clinicians found variation in style of component delivery, from 'tick-box' to patient-centred approaches. Implementation barriers included inadequate skills training to implement patient-centred care planning, but patients reported increased patient-centredness due to comprehensive reviews, extra time and being asked about their health concerns.

**Conclusions** Implementation failure contributed to lack of impact of the 3D intervention on the trial primary outcome (quality of life), but so did intervention failure since modifiable elements of intervention design were partially responsible. When a decisive distinction between implementation failure and intervention failure cannot be made, identifying potentially modifiable reasons for suboptimal implementation is important to enhance potential for impact and effectiveness of a redesigned intervention.

### Strengths and limitations of this study

► In the largest randomised controlled trial of a recommended patient-centred model of care for people with multimorbidity, we conducted a comprehensive process evaluation to examine implementation fidelity in case of a null result and to inform future implementation.

► We used mixed methods to evaluate multiple aspects of implementation and a wide range of factors that might influence implementation.

► Although distinguishing between implementation failure and intervention failure is recommended in null trials to avoid needlessly discarding a promising intervention, the distinction is difficult to apply when aspects of intervention design contribute to implementation deficiencies.

► By investigating reasons for implementation deficiencies, and distinguishing between potentially modifiable and non-modifiable reasons, we have instead provided information that is potentially more valuable than dichotomising between implementation failure and intervention failure for informing decisions about wider implementation or the need for further research.

**Trial registration number** ISRCTN06180958

## INTRODUCTION

The increasing prevalence of multimorbidity, driven by ageing populations across the world, is a major challenge to health services. There is broad consensus about how primary care for people with multimorbidity should be organised,[1–3] but little evidence about the effectiveness of recommended strategies. Reflecting this absence of evidence, the 2016 National Institute of Health and Care Excellence Multimorbidity clinical guideline recommended more research on how

Adoption by the practice – intended administrative activity
- Identify patients with ⩾ 3 long-term conditions and flag on EMIS
- Install purpose-designed electronic 3D review template
- In consultation with clinicians, allocate a named GP (and nurse if appropriate) for all reviews
- All appointments outside reviews scheduled with named GP and/or nurse and offered as longer appointments
- Schedule participating patients for 6 monthly 3D review of all conditions together in extended two-part appointments, first part with named nurse, second part with named GP
- Cancel usual long-term condition reviews, and replace with 3D review
- Run monthly monitoring searches and send them to researchers

Core components
- Continuity of care
- A comprehensive review arranged with named nurse and GP in separate appointments every six months
- Longer appointments with named GP or nurse as needed between reviews

3D multimorbidity reviews – intended GP, nurse and pharmacist activity
- Two-part long-term condition review with named nurse and GP, to address all conditions together, using new 'intelligent' 3D review template.
- Part 1 typically done by a nurse: identify patient's priorities and quality of life issues, screen for depression and complete disease checks. Create agenda for second part of review based on this information and give printed copy to patient.
- Pharmacist review of medication prior to part 2
- Part 2 typically done by a GP: address agenda, review treatment and medication adherence, aim to optimise medication and reduce treatment burden, agree health plan with patient and provide written copy
- Involvement of secondary care physician if needed

Core components
- Compile patient agenda based on patient priorities and clinical measures and provide copy to patient
- Depression screening
- Attention to quality of life
- Chronic disease monitoring
- Medication review and adherence
- Share printed health plan with actions for both patient and GP

**Figure 1** 3D intended intervention work and core components. 3D, GP, general practitioner.

best to organise primary care to address the challenge of improving care for people with multimorbidity.[3] In the largest trial to date of an intervention based on the consensus of opinion about best practice for multimorbidity care, the 3D study evaluated a patient-centred approach for the people with multimorbidity, defined for this trial as people with three or more long-term conditions on a disease registry. The approach included continuity of care and regular holistic reviews (3D reviews) in primary care (general practices in the UK) with a focus on addressing quality of life, mental as well as physical health, and polypharmacy. The hypothesis was that this would improve patient-centred care, reduce treatment burden and illness burden and improve quality of life (the trial primary outcome).[4]

Process evaluation of trials evaluating complex interventions can inform decisions about the wider implementation and applicability of those interventions. A comprehensive process evaluation can help interpret trial results and inform real-world implementation[5 6] by providing explanations when interventions are not effective.[7] This may be because of intervention failure (the intervention was delivered as intended but did not improve outcomes, so should not be implemented) and/or implementation failure (the intervention was inadequately implemented and so might need additional research to further examine effectiveness).[8] However, distinguishing implementation and intervention failure

is often not straightforward[9 10] and may require detailed examination of implementation fidelity.

We have previously reported baseline data from the 3D study,[11] main trial findings[12 13] and analysis of the patient-centredness of the 3D review.[14] At baseline, many practices had already combined multiple long-term condition reviews into one appointment but other recommended care[1 3] was less evident. For example, only 10% of patients were aware of receiving a care plan and 35% were rarely or never asked what was important to them in managing their health.[11] The main trial results showed no effect from the 3D intervention on the primary outcome of health-related quality of life (HRQOL) or other related secondary outcomes such as well-being and treatment burden, but a consistent beneficial effect on patients' experience of care as more person centred.[12] Analysis of observational and interview data about intervention delivery indicated that the main reasons for the perceived increase in patient-centredness were that when patients attended for an intervention review, they were first asked about their most important health concerns and then given a longer, comprehensive review encompassing all health issues.[14] The aim of this paper is to examine whether the measured lack of effect on the primary outcome in the 3D trial was due to implementation or intervention failure, and thereby inform future intervention development and evaluation.

## METHODS

### Setting: - the 3D study

The intervention and trial evaluation are described briefly here, having been reported in detail elsewhere.[4 12 13] The core components of the intervention included offering greater continuity of care and 6 monthly, two-part patient-centred, comprehensive health reviews, conducted by a named nurse and general practitioner (GP) and underpinned by a purpose-designed electronic template (figure 1). A pharmacist also completed an electronic medication review. Practices were expected to deliver two complete reviews to every participating patient during the trial, including all review components. However, practices could decide the detail of how they would provide the reviews, enhance continuity of care and reduce the number of review appointments. Administrators and clinicians nominated by the practices received two short (2–3 hours) training sessions from the trial team on the intervention's rationale and the use of the computer template. Online supplementary appendix 1 shows the TIDieR checklist[15] for the intervention design. Figure 1 details the work that administrative staff, clinicians and pharmacists were expected to do to deliver the intervention. Sixteen general practices received the intervention compared with 17 control practices, with 1546 participating patients.[4] However, because of staffing crises, one intervention practice stopped delivering the intervention and withdrew from the process evaluation.

### Patient and public involvement

A patient and public involvement group was set up during the development of the trial intervention to ensure that it met the perceived needs of people with multimorbidity. The group was actively involved throughout the trial in multiple ways, as reported by Mann *et al*.[16]

### Process evaluation design

The design is briefly reported here as a detailed description is provided in our earlier paper.[17] We based the design on a process evaluation framework for cluster randomised trials,[18] and also considered UK Medical Research Council guidance for process evaluation of complex interventions.[10] This, rather than qualitative methodology criteria, underpins the rigour of the research as our focus was to ensure a comprehensive process evaluation that examined all aspects of intervention implementation that might affect the results of the trial. As such, the interview schedules were semi-structured to elicit specific information to answer the process evaluation research questions and the size of our qualitative sample was determined by information power[19] regarding implementation variation and the reasons for it, rather than data saturation.

We based the process evaluation on a logic map describing the intervention design and used the logic map to inform assessment of implementation fidelity (the extent to which practices implemented the intervention as the researchers intended).[17] The assessment covered adoption of the 3D intervention (implementation of the organisational components of the intervention); delivery of 3D reviews to patients; maintenance (whether delivery is sustained over time) and reach (the number of participants who receive the intervention) (figure 2), and the important influence of context on implementation fidelity, maintenance and reach.[20–23]

### Data collection

#### Qualitative data collection in selected practices

Intervention practices were sampled at different stages for qualitative data collection (table 1). Four practices were initially purposefully sampled during early stages of the trial, using baseline data and observation of practice

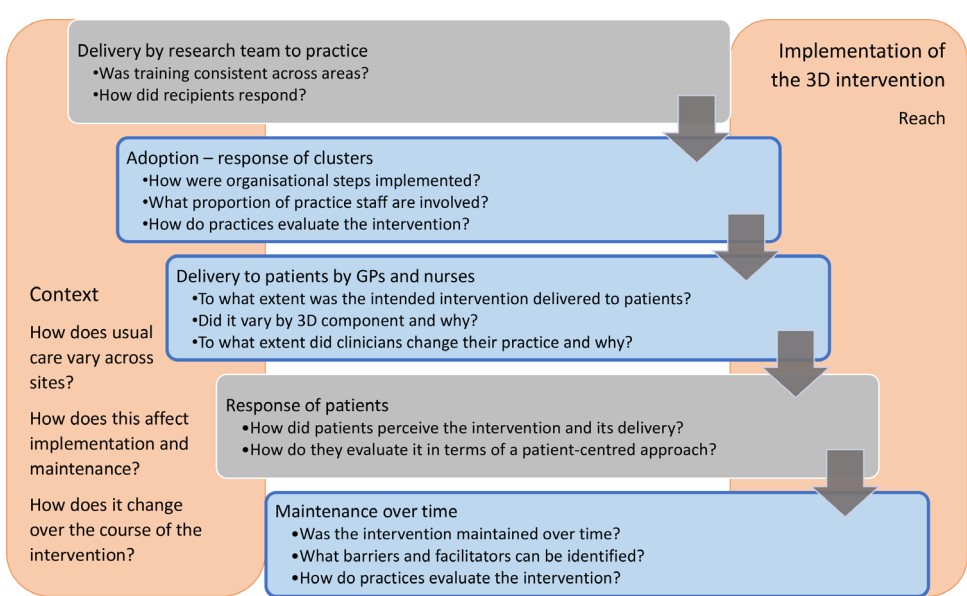

**Figure 2** Process evaluation design and research questions (research stages addressed in this paper are shown in blue). 3D; GP, general practitioner.

**Table 1** Data from intervention practices used for this study

| Data | Sampled intervention practices | Data sources | Data used to examine |
|---|---|---|---|
| Electronic data capture | All | 3D electronic template recording of reviews completed and review components delivered to all patients | Reach and maintenance Fidelity of delivery of intervention components to patients |
| Administrative survey | All | Research team completed questionnaire about organisation of reviews in all intervention practices | Adoption, reach and maintenance |
| Baseline interviews | Beddoes, Davy, Harvey, Lovell | 4 administrators, 4 nurses, 5 GPs | Individual practice context to understand adoption and reach. |
| 3D review observations | Beddoes, Davy, Harvey, Lovell, Cabot, McReady, Guppy, Carpenter | 13 nurses, 15 GPs, 22 patients[*] | Variation in delivery of intervention components to patients |
| Postreview debriefs and informal interviews | Beddoes, Davy, Harvey, Lovell, Cabot, McReady, Guppy, Carpenter | 12 nurses, 7 GPs, 10 patients | Variation in delivery of intervention components to patients Maintenance of intervention delivery |
| Patient focus groups | Beddoes, Davy, Harvey, Lovell | 22 patients† | Variation in delivery of intervention components to patients |
| End-of trial interviews | Beddoes, Davy, Harvey, Lovell, Blackwell | 4 administrators, 6 nurses, 5 GPs, 7 patients | Variation in delivery of intervention components to patients. Maintenance of intervention delivery |

*6 patients were observed for both parts of review.
†2 focus groups of 3 patients, 1 focus group of 7 patients and 1 focus group of 7 patients and 2 carers.
GP, general practitioner.

team training, for detailed qualitative investigation of all aspects of implementation, including context, adoption, delivery and maintenance. This sampling reflected our assumptions that (1) larger practices may have lower continuity of care and a lower proportion of clinicians taking part in 3D, which may influence implementation and (2) practices whose care for patients with multimorbidity already reflected aspects of the 3D approach may adopt 3D more readily. These four practices were included in every stage of data collection. An additional five practices were responsively sampled at later stages for focused observation of clinicians' style of delivery of 3D reviews and to examine variations in models of delivery that emerged during the trial. In total 9 of the 16 intervention practices were sampled.

All intervention practices were given pseudonyms to preserve anonymity. Data collected included: interviews with practice staff; non-participant observation of 3D reviews with follow-up interviews with clinicians and patients; and focus groups and interviews with patients (table 1), all of which were audio recorded. The qualitative data were almost all collected by CM, a female qualitative researcher experienced in focus groups and interviews and with clinical nursing experience, including as a practice nurse. Five observations and one interview were carried out by a female GP gaining experience in qualitative research. The interview topic guides and observation checklist are shown in online supplementary appendix 2. All the analyses were carried out by CM with support from BG, a GP, health services researcher and process evaluation methodologist, and AS, a highly experienced qualitative and process evaluation researcher.

**Interviews with practice staff:** At baseline, interviews in the four initially sampled practices with the 3D lead GP, the lead nurse and the key administrator explored usual care, initial reactions to the intervention and implementation arrangements. Interviews at the end of the trial in the same four practices, and in a fifth, responsively sampled practice where a nurse practitioner delivered all reviews, explored experience of delivering the intervention and maintenance. These interviews lasted 15–50 min. Most individuals were interviewed at both the beginning and end of the trial to achieve a longitudinal perspective on implementation and to see how their initial response to the intervention changed in light of their experience

of implementing it, but there were also a few single interviews.

**Observation of 3D reviews with follow-up interviews:** Twenty-eight 3D reviews were observed and recorded in the four initially sampled practices and in four of the responsively sampled practices, including one in which a research nurse, rather than a practice nurse, conducted most of the part 1 reviews. Observation notes were informed by an observation checklist (online supplementary appendix 2). The checklist was based on the intervention components and directed attention to whether components were delivered and the manner of their delivery. Where possible, brief follow-on interviews with the clinician and/or patient whose review had been observed were completed on the same day. These interviews lasted 5–24 min.

**Focus groups and interviews with patients**: In the four initially sampled practices, patients varying in health status and satisfaction with care according to baseline questionnaire data were invited to focus groups or individual interview towards the end of the trial, to explore their experience of receiving the intervention. One focus group per practice took place, lasting about 1 hour. Patients preferring individual interviews were interviewed for 20–50 min in a convenient location, usually their own home. All the focus groups and interviews were carried out by CM.

Online supplementary appendix 3 shows the COREQ checklist[24] for qualitative methodology and provides additional detail.

### Quantitative data collected from all intervention practices

Data about 3D review completion were extracted each month from the routine electronic medical records to evaluate intervention reach, delivery and maintenance.[4 17] The data included dates of reviews, who had completed the review, and whether core elements were recorded as delivered in the 3D review template. In the first part of the review delivered by a nurse, data included description of patients' main concerns, pain levels, depression screening, and the creation and printing of a patient agenda. The template also recorded the pharmacist's completion of a medication review, their recommendations and whether these had been noted by the GP. In the second part of the review delivered by a GP (except in one practice), recorded data included medication adherence and description of at least one main problem in the health plan, together with patient and GP actions to address the problem. Finally, the software recorded whether an agreed health plan had been printed.

### Survey data collected in all intervention practices

Researchers in each trial area completed a purpose-designed administrative survey about the way 3D reviews were organised in all intervention practices. The survey included the proportion of the administrative team involved in 3D, how patients were identified and

contacted, and whether practices facilitated 3D patients seeing their named GP at appointments other than 3D reviews.

### Data analysis

All audio recordings of qualitative data (interviews, focus groups and consultation recordings) were professionally transcribed, then the transcript was checked against the recording, anonymised and annotated with observation notes. The annotation process aided interpretation of the data and illuminated the manner of delivery in the recorded consultations. We applied qualitative description methodology to write individual accounts[25] of context and adoption of the intervention in the four practices initially sampled for detailed examination,[13] and cross-case thematic analysis[26] to identify recurring issues relevant to intervention adoption, delivery and maintenance in all nine selected practices. The data were analysed in parallel with data collection, so that emerging issues were incorporated into future data collection. For the thematic analysis, NVivo V.11 software (QSR International) was used to facilitate both deductive coding derived from intervention components and inductive coding derived from the data,[26] allowing the identification of both anticipated themes (eg, those relating to the key components of the intervention) and emergent themes across sampled practices. Qualitative analysis was led by CM with input from AS, LW and BG, who commented on the developing coding framework, double coded a sample of transcripts and agreed the final themes. Additionally, to further enhance trustworthiness and credibility of findings, two members of the patient and public involvement group each coded four transcripts to check interpretation of the data from the patient perspective. Quantitative data were analysed descriptively by CM and KC and integrated with qualitative data.

All process evaluation data collection and analyses were done blind to the trial outcome, so that interpretation would not be influenced by knowing the results of the primary outcome.

## RESULTS

The results examine (1) adoption of the intervention by practices, (2) reach and maintenance and (3) delivery of reviews to patients. In quotes, staff and patients are identified by practice pseudonym, role and a number.

### Adoption: organisational components

The two core components of organisational adoption were continuity of care and arranging the two-part 3D reviews.

### Continuity of care

Practices were asked to allocate a named GP to 3D patients for their reviews and for any appointment between reviews. Continuity of care was evaluated as a

secondary outcome for the trial and, measured using the Continuity of Care index,[27] increased slightly in the intervention arm.[12] However, some patients experienced reduced continuity because their GP left during the trial. Others were allocated a different GP for the intervention, either to share workload or because their usual GP was not participating in 3D. These patients often continued to see their usual GP for appointments other than reviews.

[My usual GP had to get changed. There's three doctors in our practice and they were doing I think 12

**Table 2** Intervention practices

| Practice | Practice size | Combined reviews at baseline* | Admin involvement | 3D review organisation | Reach | Qualitative data collection§ |
|---|---|---|---|---|---|---|
| Lovell | 4000 patients 4 GPs, 2 nurses | All combined | 1 administrator. All aware | Appointment sent, review appointments paired | First review 94% Second review 93% | In depth. All elements |
| Tothill | 10 000 patients 40 GPs, 4 nurses | Some combined | All | Appointment sent, review appointments separate | First review 92% Second review 86% | None |
| Macready | 6000 patients 6 GPs, 2 nurses | All combined | 1 administrator. All aware | Appointment sent, review appointments paired | First review 92% Second review 50% | Observation and postreview informal interview |
| Dunbar | 15 000 patients 16 GPs, 5 nurses | All combined | All | Letter inviting patient to call, review appointments paired | First review 90% Second review 75% | None |
| Cabot | 10 000 patients 12 GPs, 5 nurses | Some combined | Research nurse only | Appointment sent, review appointments separate | First review 83% Second review 74% | Observation and postreview informal interview |
| Beddoes | 5500 patients, 4 GPs, 3 nurses | All combined | All | Letter inviting patient to call, review appointments separate | First review 80% Second review 82% | In depth. All elements |
| Guppy | 8000 patients 6 GPs, 3 nurses | All combined | 1 administrator. All aware | Appointment sent, review appointments paired | First review 80% Second review 76% | Observation and postreview informal interview |
| Penn | 10 500 patients 9 GPs, 3 nurses | Some combined | 1 administrator. All aware | Phone call to patient, review appointments paired | First review 80% Second review 47% | None |
| Harvey | 15 000 patients 13 GPs, 4 nurses | Some combined | All | Appointment sent, review appointments sometimes separate | First review 77% Second review 44% | In depth. All elements |
| Priestman | 13 500 patients 10 GPs, 3 nurses | All combined | All | Letter inviting patient to call, review appointments paired | First review 75% Second review 45% | None |
| Sharples | 4500 patients 4 GPs, 2 nurses | None combined | All | Letter inviting patient to call, review appointments separate | First review 71% Second review 67% | None |
| Martineau | 5000 patients 4 GPs, 2 nurses | Some combined | 2 administrators. Others unaware | Phone call to patient, review appointments paired | First review 69% Second review 53% | None |
| Carpenter | 14 500 patients 12 GPs, 4 nurses | All combined | Unsure if all aware | Letter inviting patient to call, review appointments paired | First review 67% Second review 50% | Observation and postreview informal interview |
| Blackwell | 13 500 patients 9 GPs, 7 nurses | All combined | Nurse and administrator. Others unaware. | Letter inviting patient to call, nurse completed both parts of review | First review 66% Second review 9% | End of trial interviews |
| Davy | 14 500 patients 12 GPs five nurses | Some combined | 2 administrators. Others unaware | Appointment sent, later review appointments separate | First review 38% Second review 0% | In depth. All elements |

*Combined reviews means reviews were purposely arranged to include all long-term conditions where there was a nurse-led clinic.
†Paired means that nurse and GP appointments made at the same time but could take place on different days.
‡See table 1 for details of qualitative data collected.
GP, general practitioner.

patients, so it was split between three doctors. So I had to go with [GP2]. (Focus group Lovell Patient 8)

The four initially sampled practices (Beddoes, Davy, Harvey and Lovell) provided insight into contextual influences. Harvey already had a 'personal list' system with high continuity, but during the trial this was disrupted when several GPs left the practice. Beddoes supported 3D participants to see their allocated GP between reviews. At Davy, continuity was poorly implemented due to staff loss and because receptionists were unaware of 3D. Lovell continued with their usual system, which they felt delivered adequate continuity of care.

Most people see the doctor they want to see, so I think from a continuity point of view we know our patients very well and we've all been here a long time. [Group interview Lovell GP1]

### Arranging reviews

Administrative survey data from 15 intervention practices showed variation in the way practices arranged reviews (table 2). Ten practices involved the whole administrative team, but in four, one or two administrators arranged 3D reviews in isolation. Reach was lowest in these four practices. In the remaining practice (Cabot), a dedicated research nurse arranged all the reviews, bypassing the administrative team. Notably, some 3D patients received the 3D reviews in addition to, rather than instead of their usual individual condition reviews, as intended.

I think there became a problem where patients were being invited in for their 3D and then a couple of months later, they'd get invited in for their diabetes and their asthma because one person up there wasn't talking to the other one. [Interview Blackwell Nurse 1]

At Lovell and Harvey, existing arrangements for long-term condition reviews (one of the sampling criteria) underpinned the 3D review arrangements, reducing confusion. At Davy, the two administrators involved had to set up a different system for 3D patients. Being a large practice in which the rest of the administrative team were unaware of 3D requirements, difficulties arose when patients needed to re-arrange the appointment. At Beddoes, clinical and administrative staff decided collectively how they would implement the administrative aspects of 3D, but it differed from usual arrangements.

We'd had a team meeting after the training with the senior nurse and the GPs to decide what was the best way forward and then I met with the admin team to say, What would you like to see on your screen so that you know they're part of the 3D study and so that you know about the appointments? (Interview Beddoes practice manager)

Overall, adoption was inconsistent, affected by practices' choices in respect of continuity and arrangements

**Table 3** Quantitative evaluation of reach

| | No (%) of 3D reviews delivered |
|---|---|
| Practice level analysis | n=16 practices |
| Reach (% expected number of reviews delivered) | |
| First review | Median 66% (range 38%–94%) |
| Second review | Median 47% (range 0%–93%) |
| Patient level analysis | n=797 |
| Delivery of 3D nurse and GP reviews* | |
| Two 3D reviews with both GP and nurse (full) | 390 (49%) |
| One 3D review with both GP and nurse (partial) | 205 (26%) |
| Other (eg, nurse review but no GP review) (partial) | 31 (4%) |
| No 3D reviews (none) | 171 (21%) |

*622 (78%) patients had at least one nurse review; 599 (75%) had at least one GP review. 390 (49%) patients received a 'full' intervention (defined as having two reviews, with each review involving a nurse and a GP appointment which could be on the same day or different days that is, four appointments in total) in the 15 months of follow-up. 21% received no intervention.
GP, general practitioner.

for reviews. Duplication of reviews in some practices suggests difficulty in testing effectiveness of an intervention in a research situation that involves a short-term alteration to accustomed methods of providing care, that affects only a subset of patients.

### Reach and maintenance

Table 3 shows mean reach in all intervention practices. We defined intervention reach in terms of receipt of planned 3D reviews by participating patients. Reach varied between practices from 38% and 94% (median 66%) of all recruited patients in a practice receiving both the nurse and GP appointments in first round reviews, and between 0% and 93% (median 47%) in second round reviews. Initial implementation of the intervention was, therefore, not well-maintained.

In the four initially-sampled practices, the qualitative data revealed contextual factors reducing the time window for delivering reviews. Lovell started delivering 3D reviews straight after training and had the highest reach of any practice in the intervention arm. The other three practices delayed starting, Davy because of the sudden loss of three of their long-term condition nurses and two GPs, Harvey because they were changing their system for sending letters re-calling patients for long-term reviews, and Beddoes because of staff sickness. Once started, Davy administrators struggled to organise reviews, hampered by ongoing sickness in the nursing team, and only

managed to schedule 25% of the reviews required. The greatest challenge was accommodating paired reviews within over-stretched appointment schedules.

> And I think because you're trying to tally it up with the doctor and the nurse, trying to find the time with the nurse if they've got more than one problem … and again they're not full time; they work part time. [Interview Davy Administrator 1]

Difficulties with arranging appointments reinforced practices' initial fears that the time demand and workload of implementing the 3D intervention would be too great. One suggestion made by GPs was that patients could be selected using more stringent criteria to reduce the overall number and maximise the chance of benefit. Another suggestion, from nurses, GPs and patients, was that the reviews need not involve the GP every time and/or could be shorter. Some comments suggested a lack of perceived value of the second-round reviews and that a second-round review with the nurse alone would be more time-efficient.

I know they need to be reviewed but do they need to be reviewed by nurse and GP?

> … because if we saw them for review and they were happy. Do they honestly need to see the GP to say "Are you still happy, like from last week"? (Interview Guppy Nurse 1)

Practices may therefore have been less motivated to arrange second reviews, and one practice reported that fewer patients responded to the invitation to attend them.

> As a practice we've actually struggled to get them in for their second ones … we've written to them all twice – probably 30% of them haven't booked in and

so we have had a bigger DNA rate for the second ones than the first ones. (Interview Beddoes GP1)

Overall, reach and maintenance were lower than intended, indicating a degree of implementation failure. Attention to context showed this was mainly a result of unanticipated events (eg, staff loss or sickness) affecting practice capacity. However, aspects of intervention design (eg, the inclusion of two reviews in 1 year with both nurse and GP each time) may also have impacted reach and maintenance.

### Delivery of 3D review components

In 3D reviews that took place, each of the intervention components (see figure 1) detected by the electronic search were completed in at least 92% of the delivered reviews, except medication adherence which was completed in 84% and printing the health plan in 77% (table 4 and online supplementary appendix 4). The qualitative data provided insight into reasons for less consistently recorded components but also found evidence of significant variation in the manner of delivery suggesting that the high recorded component completion concealed some tick-box compliance. Variation in the patient-centredness of review component delivery has been reported in more detail in a previous paper[14]; here we focus primarily on implementation fidelity.

### Eliciting and documenting the patient's concerns (most important problem noted)

The most consistently delivered component (99% completion) (table 4), was asking patients about the health problems important to them. Nurses often invited disclosure of all health concerns, large or small.

| Table 4 | Quantitative evaluation of component delivery |
| --- | --- |
| | **No (%) of each element of the 3D review delivered** |
| Delivery of pharmacist medication review | 607/797 (76) |
| For those with at least one GP or nurse review | |
| Most important problem noted (patient agenda)* | 616/622 (99) |
| EQ-5D pain question noted (quality of life)* | 611/622 (98) |
| PHQ-9 depression screening noted* | 599/622 (96) |
| Patient agenda printed* | 579/622 (93) |
| Medication adherence noted† | 506/599 (84) |
| First patient problem noted† | 590/599 (98) |
| Noted 'what patient can do' for first problem (health plan)† | |
| Noted 'what GP can do' for first problem (health plan)† | 559/599 (93) |
| 3D health plan printed† | 461/599 (77) |

*Components delivered in the nurse part of the review of which 622 took place. If one patient had two reviews, this component was delivered in at least one.
†Components delivered in the GP part of the review of which 599 took place. If one patient had two reviews, this component was delivered in at least one.
GP, general practitioner.

She said to me, 'Is there anything you want to discuss with me at all, anything?' [Focus group Beddoes Patient 4)]

Some GPs and nurses commented on the value and novelty of asking about all patients' health concerns at the start of the consultation[14] but others were conscious of their clinical responsibility for managing the long-term conditions. Therefore, they preferred to separate the long-term conditions from health concerns they viewed as more trivial, or disabilities not amenable to change.

They want to discuss … the things that are happening to them at that particular moment … they've got a bad cold, or the cat's died or something else and they don't want to talk about their diabetes or their COPD. [Interview Beddoes GP3]

There was also observed variation in how patient's concerns were elicited, recorded in the agenda and addressed in the health plan. The printed agenda was intended to reflect the patient's perception of health problems (as well as clinical concerns), but nurses were often observed to reframe patients' problems into more medical terms. For example, one patient said: 'I can't take these naproxen now because … they've upset my stomach' and the nurse recorded 'gastric problems'. This medicalisation of problems may have contributed to some patients' perception that the agenda was simply a means for the nurse to communicate their findings to the GP, rather than an agenda that the patient owned.

They just went through everything, all the problems, the nurse did and just wrote this report out for [GP2]. [Focus group Beddoes Patient 11]

### Quality of life and depression screening

Although completion was high, observation revealed that components that had a range of set answers were sometimes delivered in a 'tick-box' way that did not invite dialogue. This most commonly happened with template questions about quality of life and depression screening. It usually occurred when the nurse anticipated no problems being revealed but in interview some nurses also said that they lacked confidence in talking to patients about mental health.

### Printing patient agenda

The patient agenda was printed in the vast majority of cases (93%) (table 4) but problems with printing were occasionally observed and one nurse said she asked patients if they wanted it and that they declined. This may have reflected a perceived lack of ownership of the agenda by the patient.

Would you like a copy? And they're like, it's fine… Nobody has wanted a copy. [Interview Davy Nurse 1]

### Medication adherence

The completion rate of this component was lower at 84% but the qualitative data did not reveal why, other than some GPs' preference to complete the template after the review, which may have meant they forgot to ask about it. On the contrary, there was evidence of some support for this component among GPs.

I do think the thing about tablets that patients take and which ones they don't like, if any, is useful. [Interview Lovell GP1]

### Collaboratively agreeing a plan

Health plans were intended as collaborative agreements between patient and GP, recording identified problems and specific actions for patient and GP to address each recorded problem. The patient and GP actions were well completed (93% and 92%, respectively, for the first problem) but the health plan was printed less frequently (77%) (table 4). This may reflect GPs apparent dislike of the health plan and a perceived lack of value, as well as observed technical difficulties printing the plan. Interview data included reservations about the formulation of the health plan, which may have made GPs reluctant to give them to patients.

I felt it was almost that you were actually chiding them in some ways, to say, 'You should do this, should do that. … It's almost like when we were at primary school, taking home your homework tasks and goals for the week'. [Group interview Lovell GP3]

During observations, a collaborative dialogue based on patients' chosen goals was seldom generated, and most plans were based on actions suggested by the GP. Some GPs commented that patients had not given prior thought to what they wished to address and that sometimes it was difficult to identify problems to include in the plan.

That's where I think perhaps them thinking in advance about their goal setting would help aid the conversation because often they say 'No, no there's nothing I want to discuss' and you eventually tease out one or two things from them. [Interview Beddoes GP1]

Some clinicians felt that the training provided by the trial team was insufficient to enhance skills required for agenda setting and especially collaborative action-planning.

I think some kind of communication training … would have been useful…there was a little bit about goal setting and confidence skills but there was no real practical element to it so in some ways you're testing what we already do but in a different context. [Interview Lovell GP1]

Others would have liked some training follow-up to check if they were delivering the intervention as intended,

and additional training prior to the second round of reviews to ensure they were 'doing it right'.

In conclusion, although the quantitative data indicated that the intervention components were delivered for a high proportion of patients receiving reviews, the qualitative data showed that delivery style varied in ways that could sometimes compromise their function. Some components, such as creating the health plan, could have benefited from more training.

## DISCUSSION
### Summary of findings
The process evaluation identified that implementation was somewhat deficient in adoption (arranging the requisite number of 3D reviews, ensuring continuity of care, reducing the overall number of reviews) and aspects of delivery (creating health plans), but most delivered reviews included all components. Reasons for incomplete implementation included unexpected pressure on resources, implementation choices made by practices (including not involving the entire administrative team), and insufficient training for using patient-centred approaches. During delivery of reviews to patients, using the template was the key to maintaining 'fidelity of form', but variation in the patient-centredness of delivery sometimes undermined 'fidelity of function'.[28] The overall prediction made by the process evaluation team while blind to the trial results was that the intervention would have improved patient experience in patients who received 3D reviews, but not changed HRQOL (the findings were presented and this prediction made at the trial steering committee meeting immediately before unblinding). The prediction of improved experience was based on the positive feedback from patients in focus groups and interviews suggesting improvements in their perceptions of care. The prediction of unchanged HRQOL was based on limited engagement of patients in the health plans (observed and described by clinicians), a lack of evidence of major changes to quality of care and feedback from administrators and clinicians about difficulties organising reviews. The trial results confirmed these predictions,[12] which increase our confidence in the process evaluation findings.

### Strengths and weaknesses
Strengths include predesigning the process evaluation based on a published framework for process evaluation of cluster randomised trials[10 17 18] covering all trial stages and maintaining responsiveness to emerging information. This maximised the likelihood that all factors that might influence implementation fidelity, including context, were considered.[7] Data of varying and complementary types were collected from a wide range of sources, both purposively sampled and cross-trial. The purposive sampling of practices mitigated the limitation that only a subset of practices and individuals involved in the trial were interviewed or observed, and we explored the full range of variation in implementation and reach (table 2), including quantitative process data from all practices. In accordance with published guidance,[10] the process evaluation analysis took place blind to the trial results.

### Comparison to other literature
An aim of the 3D process evaluation was to examine implementation fidelity to distinguish between implementation failure and intervention failure in the event of a null result. This distinction matters because it is important to avoid discarding a potentially effective intervention that was poorly implemented.[10 29 30] Implementation difficulties and deficiencies are not infrequently identified in effectiveness evaluations of complex healthcare delivery interventions[31–34] but are not always elucidated.[20 35] In this study, we found evidence of a degree of implementation failure and, in addition to identifying poorly implemented components, we have considered reasons for poor implementation and whether they are modifiable. Non-modifiable reasons include unexpected events in individual practices, most commonly staff leaving and not being easily replaceable. Potentially modifiable reasons for adoption problems include the individual choices practices made about arranging reviews, influenced by practice size and existing recall systems, but implementation was also affected by the research trial context. Implementation in these circumstances is short term, and only applies to a subset of patients, with the majority still receiving usual care, which increases the risk of confusion and duplication. This circumstance influenced administrative choices made by practices, which in turn affected implementation.

The role of intervention design and setup, including training provided by research teams to practices, is significant and modifiable. In common with other research teams, we experienced difficulty in establishing a new way of working,[36 37] although care did change enough that patients reported statistically significant changes in their experience of care in the intended direction (eg, having a greater sense of being consulted about their experience of health) and greater satisfaction with their care.[12] The evidence suggested that this was attributable to the design of the intervention reviews (longer, comprehensive and asking first about the patient's concerns),[14] but there was also evidence that intervention design negatively affected implementation in some potentially modifiable ways. Implementation of health plans suffered from insufficient training and a lack of coherence between the health plan format and GP current practice, clearly suggesting that intervention design relating to both these aspects could be improved. Professional perceptions that some patients were unprepared to engage in health planning suggests that additional patient-targeted intervention components and/or better clinician training addressing attitudes and barriers to engaging in health planning and supporting self-management[38] might facilitate collaboratively agreeing a plan of action. Many professionals did not see value for many patients in doing a second comprehensive

review in the same year, which likely contributed to lower reach for second reviews, and suggests that more targeted follow-up might have been a better design than routine rereview for all.

Our overall judgement was that there was, therefore, evidence of both implementation failure and intervention failure, but that these were linked rather than truly distinct because in this case aspects of intervention design influenced implementation. Improvements in intervention design could be focused on incorporating skills practice in the 3D training, better selection and preparation of patients, improvement to the health plan including a different format and greater patient ownership. We could also consider greater flexibility in follow-up reviews to allow varying intensity of follow-up tailored to patient need.

There is, however, a dilemma between ensuring an intervention is implemented with high fidelity and allowing flexibility to suit local circumstances. The intervention design did allow for some adaptation 'at the periphery'[39] and distinguished between core components that must be implemented in a particular form and less closely specified components whose form could vary, as long as the intended function was achieved.[28] This is recommended to facilitate implementation in individual practices, but it is not straightforward to choose where to specify intervention elements as 'central' and where to allow flexibility. In retrospect, some flexibility in follow-up reviews would be reasonable in future iterations of this type of intervention. A further change, which might plausibly alter impact on HRQOL, would be to evaluate implementation over a longer period (although that clearly has significant cost implications) or as a whole practice improvement intervention delivered to all eligible patients, rather than running a parallel system of care for individual trial participants. However, this creates the paradox that providing an intervention outside the context of a research trial may be more likely to provide a true representation of its effectiveness, but the effectiveness cannot be proved without the research.

## CONCLUSIONS

In the context of an intervention that followed the recommendations and best evidence for care of people with multimorbidity, where the trial provided strong evidence that there was no effect on the primary outcome of HRQOL but an improvement in patient-centred outcomes, we found evidence of both implementation and intervention failure. Although this challenges the assumption that implementation and intervention failure can be clearly distinguished, we believe that the distinction does provide a useful framework to help interpret trial findings and to systematically identify modifiable and non-modifiable factors to inform future implementation decisions. This paper provides a worked example of how to use these concepts in process evaluation. We conclude first, that in the case of the 3D trial a truer test of the

intervention effectiveness might be achieved by modifications that support better implementation, including whole practice implementation over a longer period to allow embedding. Second, it is important to examine reasons for implementation deficiencies to determine whether there were implementation failures and the reasons for them and whether they might be modifiable in order to avoid discarding a potentially effective intervention.

**Acknowledgements** The authors sincerely thank all those practices, staff and patients who took part in the 3D trial and especially those who contributed to the process evaluation by consenting to interviews and observation. We thank Sue Cooke and Polly Duncan who collected and helped with the analysis of some of the qualitative data in the lead trial site, Rebecca Robinson who supported the process evaluation with data collection in the early stages, Daisy Gaunt who did the main quantitative analysis for the trial, and the researchers in supporting trial sites, Bridie FitzPatrick, John McLeod, Caroline Gardner, Victoria Lee and Keith Moffat who supported the process evaluation by facilitating data collection and collecting administrative survey data. We also thank Pete Bower and Stewart Mercer, the principal investigators at those sites, who contributed to trial team discussions on the process evaluation design and findings. We also acknowledge other members of the wider trial team, Sandra Hollinghurst, Bryar Kadir, Emma Moody, Imran Rafi and Joanna Thorn. Finally, we thank the patient and public involvement group who contributed valuable insight into qualitative findings and great enthusiasm for the study.

**Contributors** CM, ARGS and BG designed the process evaluation. CS led the design of the 3D intervention and the randomised trial. CM collected and analysed the qualitative data with input from ARGS, LW and BG and led the analysis and write-up of the results presented in this paper. ARGS, BG and CS critically revised the manuscript. KC helped to design the template, analysed the quantitative data it recorded and helped to collect administrative survey data. M-SM contributed to the design of the process evaluation and facilitated data collection in the role of trial manager. All authors discussed findings, commented on the paper and approved the final version.

**Funding** This project was funded by the National Institute for Health Research Health Services and Delivery Research Programme (project number 12/130/15).

**Disclaimer** The views expressed are those of the author(s) and not necessarily those of the NIHR or the Department of Health and Social Care.

**Competing interests** None declared.

**Patient consent for publication** Not required.

**Ethics approval** The trial and process evaluation were approved by the South-West England NHS Research Ethics Committee (14/SW/0011)

**Provenance and peer review** Not commissioned; externally peer reviewed.

**Data availability statement** Data relevant to the study are included in the article or uploaded as online supplementary information.Some qualitative data are subject to restricted access and may be available only on application to the author

**ORCID iDs**
Cindy Mann http://orcid.org/0000-0002-1256-1820
Mei-See Man http://orcid.org/0000-0003-4948-5670
Katherine Chaplin http://orcid.org/0000-0003-1261-9938
Chris Salisbury http://orcid.org/0000-0002-4378-3960

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
