## [Reviewer comments · BMJ Open]

ARTICLE DETAILS

TITLE (PROVISIONAL)	Can implementation failure or intervention failure explain the result of the 3D multimorbidity trial in general practice: mixed methods process evaluation
AUTHORS	Mann, Cindy; Shaw, Ali; Guthrie, Bruce; Wye, Lesley; Man, Mei-See; Chaplin, Katherine; Salisbury, Chris

VERSION 1 – REVIEW

REVIEWER	Maxime Sasseville Université du Québec à Chicoutimi, Canada
REVIEW RETURNED	08-Jul-2019

GENERAL COMMENTS	The authors tackle a very important topic in understanding the efficacy of multimorbidity interventions. The paper is clear and very well written, however, because of the tremendous task of collecting and analyzing multiple data sources, the paper lacks details on how this was achieved. Minor comments about these issues are provided below, I hope that my comments will help the authors convey their message. Abstract: L27-31: The conclusion are quite vague and mostly about the process. I think the abstract would benefit from a more impactful message. Introduction: Could the authors give precision on the definition of multimorbidity that was used for the trial and consequently this paper, I think it would help uninitiated readers. L75-76: The statement about distinguishing the two types of failure falls flat, would it be interesting how you will be positioning the paper regarding the difficulty of distinction or link it with the following paragraph? L88-91: The aim of the study used the verb observed, but it was a lack of difference in means of a measurement. L88-91: The aim statement lacks clarity, mostly because of the « , with a view to interpreting trial findings, ». A reformulation would help the clarity of the aim. Process Evaluation Design L114-122: It think it would help if the authors clarified the differences between their use of qualitative inquiry in a context of
---

	process evaluation in comparison with a classical approach of qualitative description. I think it would help readers navigate the paper without searching for rigour criteria like data saturation, for example. I know that it is in the checklist in appendix 2 but I was not sufficient for me to answer my questions. I would like to add that this problem may also emanate from a too short description of the process evaluation method and that by clarifying the design section it would resolve this issue. L145-146: Discrepancies were identified between the text identifying time of interviews from 15-50 and 5-50 in appendix 2. If 5-50 is the right time, could you please add a sentence in the discussion about the very large difference in time. L145-146: Could you justify why you interviewed the same participants twice? It was identified that content validation was not used in this project in appendix 2. It is usually not valid to return to the same participants with the same interview guide as it introduces bias in the data collection process. L147-150: I think that it should be stated that an observation list was used (appendix 2), and also how that list was developed and updated if needed. L141-156: I think that the data collection information should include who did the interview and observation and who directed the focus group and their experience in interviewing or focus group direction. The authors argue that in point 8 of the appendix 2 that this information is irrelevant (or not as important) for implementation evaluation. I would like them to add precision to back that statement, because for me, any source of bias in a qualitative data collection process will have an effect on the result. (Linked with comment L114-122) L141-156: The guides and observation list could be provided as an appendix to help convey the quality of the process. L178: The author indicates that the qualitative description was elaborated using reference 25 an opinion piece by Sandelowski about the state of qualitative research. Could the authors identify what was the methodology paper or book used to guide the qualitative method.? L179: The author indicates a cross-case analysis, but it was never mentioned that a multiple case study method was used in this analysis. Could the authors add precision on what the mean? L176-193: The analysis of the observation data is absent from the paragraph, could the authors add precision on the process data analysis and integration. Results: L323-328: Do the authors have a reason or an exploration behind the lack of need behind having a copy of the agenda? Discussion and Conclusion: No further comments, discussion is in line with results.
--	---

REVIEWER	Gill Harvey University of Adelaide, Australia
REVIEW RETURNED	11-Jul-2019

GENERAL COMMENTS	Thank you for the opportunity to review this paper, which I enjoyed reading. It is well written and gives a clear and comprehensive account of a detailed process evaluation embedded within a trial of a complex intervention. The paper will be a valuable read for other researchers undertaking similar complex studies within the real-world setting, particularly in terms of understanding the issues of intervention-implementation success or failure. I have just a number of minor queries and suggestions that I think would be useful to address:  - On page 4, Introduction, lines 59 -61. I found it confusing that the 2nd sentence of this paragraph indicates that the NICE guideline recommended more research on how best to organise primary care to address the challenges presented by multimorbidity. Then the following sentence states that "There is broad consensus about how primary care for people with multimorbidity should be organised". This seems to contradict the previous sentence. - In the methods section, the core components of the intervention are described, as are aspects where the practices could decide details of the delivery at a practice level. This concept of core components and an adaptable periphery is described in implementation frameworks such as CFIR (Consolidated Framework for Implementation Research) and might be worth referencing or returning to later on in the discussion. - Page 7 data collection. I found the description of sampling difficult to follow in parts. The initial sampling of 4 practices is described as purposive and appears to be related to criteria such as practice size and baseline data about patient review. This could be made clearer within the paper. Similarly, the later sampling of 5 practices is described as 'responsive'. What does this mean? I was also unclear whether it was the same 4 practices plus an additional one that were sampled for the second round of the process evaluation data collection. In the introduction, the authors state that 9 out of 16 practices were selected for qualitative data collection; however, in the description of interviews with practice staff (line 141 -146), it seems that interviews were repeated in the initial 4 practices plus one additional one, which seems to suggest only 5 distinct practices were included in the process evaluation. Table 1 did not really help to clarify this. It would be helpful to identify (maybe in Table 1 or 2) exactly which practices were part of the process evaluation. - Contextual factors were identified as important and there is discussion of these in terms of staff turnover and sickness etc. It would be interesting to know if other contextual factors such as practice size, geographical location, policy environment etc. had any effect on the implementation of the intervention. - In the discussion section, lines 377-382, the authors refer to their prediction of the trial results. This is interesting, but it would be good to have seen it unpacked a bit further. What aspects of the process evaluation led them to make this prediction? - In the summing up and conclusions, I really liked the discussion around the links between intervention and implementation failure. This is a helpful contribution to the literature on implementation research.
--

VERSION 1 – AUTHOR RESPONSE

Reviewer 1

The authors tackle a very important topic in understanding the efficacy of multimorbidity interventions. The paper is clear and very well written, however, because of the tremendous task of collecting and analyzing multiple data sources, the paper lacks details on how this was achieved.

Response: Thank you for taking the time to review this paper and for your useful detailed comments. We have added detail in various ways you have suggested into the main paper, and made it clear that additional detail is also to be found in the COREQ (Appendix 3) NB: Please note that the appendices have been re-numbered at the request of the journal so the COREQ is now Appendix 3

Abstract:

L27-31: The conclusion are quite vague and mostly about the process. I think the abstract would benefit from a more impactful message.

Response: Thank you for suggesting the impact of the abstract conclusion could be greater. The final sentence has been changed to: 'When a decisive distinction between implementation failure and intervention failure cannot be made, identifying potentially modifiable reasons for sub-optimal implementation is important to enhance potential for impact and effectiveness of a re-designed intervention'.

Introduction:

Could the authors give precision on the definition of multimorbidity that was used for the trial and consequently this paper, I think it would help uninitiated readers.

Response: The definition of multimorbidity used for the trial (those with 3 or more long-term conditions on a disease registry) has been added in the first paragraph at line 68

L75-76: The statement about distinguishing the two types of failure falls flat, would it be interesting how you will be positioning the paper regarding the difficulty of distinction or link it with the following paragraph?

Response: We have tried to clarify the paragraph about the distinction between intervention failure and implementation failure and its significance and linked it better to the following paragraph by adding 'and may require detailed examination of implementation fidelity' to the last sentence at line 81-82

L88-91: The aim of the study used the verb observed, but it was a lack of difference in means of a measurement.

Response: Thank you for pointing out this inaccuracy. We have changed 'observed' to 'measured' Line 95

L88-91: The aim statement lacks clarity, mostly because of the « , with a view to interpreting trial findings, ». A reformulation would help the clarity of the aim.

Response: The aim has been clarified to read: 'The aim of this paper is to examine whether the measured lack of effect on the primary outcome in the 3D trial was due to implementation or intervention failure, and thereby inform future intervention development and evaluation' Line 94-97

Process evaluation design:

L114-122: It think it would help if the authors clarified the differences between their use of qualitative inquiry in a context of process evaluation in comparison with a classical approach of qualitative description. I think it would help readers navigate the paper without searching for rigour criteria like data saturation, for example. I know that it is in the checklist in appendix 2 but I was not sufficient for me to answer my questions. I would like to add that this problem may also emanate from a too short description of the process evaluation method and that by clarifying the design section it would resolve this issue.

Response: We have considered your comments about the lack of detail and have added detail where specifically requested. We have also added a new section (Lines 123-128) to the first paragraph of

the process evaluation design description and added a new reference about information power to show better how the research aims of a process evaluation are likely to result in a different approach from the standard qualitative descriptive one. To preserve readability and after discussing with the editor the acceptability of providing instead a detailed description in the COREQ form in Appendix 3, we have kept it there. We have also referred to the process evaluation protocol published in BMJ Open in 2016 which contains the full detail of the design.

L145-146: Discrepancies were identified between the text identifying time of interviews from 15-50 and 5-50 in appendix 2. If 5-50 is the right time, could you please add a sentence in the discussion about the very large difference in time.

L145-146: Could you justify why you interviewed the same participants twice? It was identified that content validation was not used in this project in appendix 2. It is usually not valid to return to the same participants with the same interview guide as it introduces bias in the data collection process.

Response: We have now differentiated between the time taken for formal interviews (15-50 minutes) and brief interviews following review observations (5-24 minutes) and explained why some participants were interviewed twice, which is usual practice in a longitudinal study. Lines 163-167

L147-150: I think that it should be stated that an observation list was used (appendix 2), and also how that list was developed and updated if needed.

Response: We have added that an observation checklist was used at line 171 and how it was designed. This is included in new Appendix 2 along with the interview topic guides. Lines 171-173

L141-156: I think that the data collection information should include who did the interview and observation and who directed the focus group and their experience in interviewing or focus group direction. The authors argue that in point 8 of the appendix 2 that this information is irrelevant (or not as important) for implementation evaluation. I would like them to add precision to back that statement, because for me, any source of bias in a qualitative data collection process will have an effect on the result. (Linked with comment L114-122)

Response: This information has now been added at Lines 152-158. We agree with your observation that the statement in point 8 of Appendix 3 is inadequately justified and have removed it. Some information about the researcher characteristics has now been included in the paper itself as requested and more detail can be found in the COREQ

L141-156: The guides and observation list could be provided as an appendix to help convey the quality of the process.

Response: The observation checklist and interview guides have been included in Appendix 4 referenced at Line 155

L178: The author indicates that the qualitative description was elaborated using reference 25 an opinion piece by Sandelowski about the state of qualitative research. Could the authors identify what was the methodology paper or book used to guide the qualitative method?

L179: The author indicates a cross-case analysis, but it was never mentioned that a multiple case study method was used in this analysis. Could the authors add precision on what the mean?

Response: Reference 25 is to a paper by Sandelowski describing and justifying the methodology of qualitative description. We applied this method to the data regarding context and adoption collected from the 4 initially sampled practices to produce individual descriptions of each of the 4 practices that related to context and adoption and used thematic analysis as described in the paper indicated by reference 26 to identify themes that ran across all the sampled practices relating to adoption, delivery and maintenance. We have clarified this distinction in lines 205-209

L176-193: The analysis of the observation data is absent from the paragraph, could the authors add precision on the process data analysis and integration.

Response: The observation data were integrated by being annotated into the transcripts of the consultation recordings and therefore informed interpretation of those data by illuminating manner of delivery. We have added a sentence to this effect at lines 204-205.

Results:

L323-328: Do the authors have a reason or an exploration behind the lack of need behind having a copy of the agenda?

Response: The hypothesis was that sometimes patients did not perceive the agenda as being really theirs and so did not always value it. We have added words to this effect at line 354

Reviewer 2

Thank you for the opportunity to review this paper, which I enjoyed reading. It is well written and gives a clear and comprehensive account of a detailed process evaluation embedded within a trial of a complex intervention. The paper will be a valuable read for other researchers undertaking similar complex studies within the real-world setting, particularly in terms of understanding the issues of intervention-implementation success or failure.

Response: Thank you for your helpful comments and highlighting the value of the paper to other researchers facing similar issues of intervention/implementation success or failure.

On page 4, Introduction, lines 59 -61. I found it confusing that the 2nd sentence of this paragraph indicates that the NICE guideline recommended more research on how best to organise primary care to address the challenges presented by multimorbidity. Then the following sentence states that "There is broad consensus about how primary care for people with multimorbidity should be organised". This seems to contradict the previous sentence.

Response: We have changed the wording and reversed the order of the sentences in the first paragraph to make it clearer that although there is consensus of opinion there is a lack of evidence, hence the NICE recommendation. Lines 61-65

In the methods section, the core components of the intervention are described, as are aspects where the practices could decide details of the delivery at a practice level. This concept of core components and an adaptable periphery is described in implementation frameworks such as CFIR (Consolidated Framework for Implementation Research) and might be worth referencing or returning to later on in the discussion.

Response: Methods section: thank you for the suggestion about CFIR. A paragraph has been added to the discussion including this reference and the surrounding text has been changed to integrate this into the discussion. Lines 468-475

Page 7 data collection. I found the description of sampling difficult to follow in parts. The initial sampling of 4 practices is described as purposive and appears to be related to criteria such as practice size and baseline data about patient review. This could be made clearer within the paper. Similarly, the later sampling of 5 practices is described as 'responsive'. What does this mean? I was also unclear whether it was the same 4 practices plus an additional one that were sampled for the second round of the process evaluation data collection. In the introduction, the authors state that 9 out of 16 practices were selected for qualitative data collection; however, in the description of interviews with practice staff (line 141 -146), it seems that interviews were repeated in the initial 4 practices plus one additional one, which seems to suggest only 5 distinct practices were included in the process evaluation. Table 1 did not really help to clarify this. It would be helpful to identify (maybe in Table 1 or 2) exactly which practices were part of the process evaluation.

Response: We have clarified the description of the sampling and also added information to Table 1 (the practice pseudonyms) to make it clear which practices were involved at each stage of data collection. Lines 138-148

Contextual factors were identified as important and there is discussion of these in terms of staff turnover and sickness etc. It would be interesting to know if other contextual factors such as practice size, geographical location, policy environment etc. had any effect on the implementation of the intervention.

Response: Contextual factors: we have added some words to the discussion section at line 438 to highlight that the contextual influences included existing organisational arrangements and size of practice as reflected in the data (Lines 241-245 and 260-266)

In the discussion section, lines 377-382, the authors refer to their prediction of the trial results. This is interesting, but it would be good to have seen it unpacked a bit further. What aspects of the process evaluation led them to make this prediction?

Response: We have added some information at lines 409-414 to explain the basis for these predictions

In the summing up and conclusions, I really liked the discussion around the links between intervention and implementation failure. This is a helpful contribution to the literature on implementation research.

VERSION 2 – REVIEW

REVIEWER	Maxime Sasseville Université du Québec à Chicoutimi Canada
REVIEW RETURNED	12-Aug-2019

GENERAL COMMENTS	I would like to congratulate the authors for the tremendous work done in answering difficult comments in the first round of reviews. I think that the changes improve the clarity of the methods to support already very well described results. My comments were answered fully and I have no further comments.
--

REVIEWER	Gill Harvey University of Adelaide, Australia
REVIEW RETURNED	21-Aug-2019

GENERAL COMMENTS	Thank you for the opportunity to review the revised manuscript. I am happy that the authors have responded appropriately to my initial comments, particularly the additional information provided in the methods.
---